# Older Fallers’ Comprehensive Neuromuscular and Kinematic Alterations in Reactive Balance Control: Indicators of Balance Decline or Compensation? A Pilot Study

**DOI:** 10.3390/bioengineering12010066

**Published:** 2025-01-14

**Authors:** Ringo Tang-Long Zhu, Timmi Tim Mei Hung, Freddy Man Hin Lam, Jun-Zhe Li, Yu-Yan Luo, Jingting Sun, Shujun Wang, Christina Zong-Hao Ma

**Affiliations:** 1Department of Biomedical Engineering, The Hong Kong Polytechnic University, Hong Kong SAR 999077, China; ringo-tanglong.zhu@connect.polyu.hk (R.T.-L.Z.); timmei-timmi.hung@connect.polyu.hk (T.T.M.H.); junzhe.li@connect.polyu.hk (J.-Z.L.); yuyan-laura.luo@connect.polyu.hk (Y.-Y.L.); shu-jun.wang@polyu.edu.hk (S.W.); 2Research Institute for Smart Ageing, The Hong Kong Polytechnic University, Hong Kong SAR 999077, China; 3Department of Rehabilitation Sciences, The Hong Kong Polytechnic University, Hong Kong SAR 999077, China; freddy-mh.lam@polyu.edu.hk; 4Future Architecture and Urban Research Institute, Tongji Architectural Design (Group) Co., Ltd., Shanghai 200092, China; 3002_sunny@tongji.edu.cn

**Keywords:** community-dwelling, older adults, falls, reactive balance, perturbation, electromyographic (EMG), co-contraction index (CCI), kinematics, postural sways

## Abstract

**Background**: Falls and fall consequences in older adults are global health issues. Previous studies have compared postural sways or stepping strategies between older adults with and without fall histories to identify factors associated with falls. However, more in-depth neuromuscular/kinematic mechanisms have remained unclear. This study aimed to comprehensively investigate muscle activities and joint kinematics during reactive balance control in older adults with different fall histories. **Methods**: This pilot observational study recruited six community-dwelling older fallers (≥1 fall in past one year) and six older non-fallers, who received unpredictable translational balance perturbations in randomized directions and intensities during standing. The whole-body center-of-mass (COM) displacements, eight dominant-leg joint motions and muscle electrical activities were collected, and analyzed using the temporal and amplitude parameters. **Results**: Compared to non-fallers, fallers had significantly: (a) smaller activation rate of the ankle dorsiflexor, delayed activation of the hip flexor/extensor, larger activation rate of the knee flexor, and smaller agonist-antagonist co-contraction in lower-limb muscles; (b) larger knee/hip flexion angles, longer ankle dorsiflexion duration, and delayed timing of recovery in joint motions; and (c) earlier downward COM displacements and larger anteroposterior overshooting COM displacements following unpredictable perturbations (*p* < 0.05). **Conclusions**: Compared to non-fallers, fallers used more suspensory strategies for reactive standing balance, which compensated for inadequate ankle/hip strategies but resulted in prolonged recovery. A further longitudinal study with a larger sample is still needed to examine the diagnostic accuracies and training values of these identified neuromuscular/kinematic factors in differentiating fall risks and preventing future falls of older people, respectively.

## 1. Introduction

Falls and fall consequences in older adults burden society heavily and are global health issues [1]. Annually, around one in three older adults falls, one in ten older adults has fall-related injuries, and 684,000 fall-related deaths happen worldwide [1,2]. However, even the multi-factorial fall-prevention management has shown relatively limited success in fall reduction, especially in older adults with fall histories, i.e., fallers [3]. Given that balance and gait disorders are the second leading causes of falls except accidents [1], some in-depth physiological alterations of balance control in older fallers that have remained unidentified may be modifiable to prevent older adults’ future falls more effectively.

Several balance control strategies with the involvement of lower limbs have been proposed, based on the analyses of kinematics (i.e., postural sways, joint motions) and neuromuscular activities (i.e., electromyographic [EMG] signals). The feet-in-place strategy is commonly employed to keep the whole-body center of mass (COM) within the base of support (BOS) when external perturbations are not large; this comprises a single or a combination of the ankle strategy, hip strategy, and suspensory strategy (bending knees to lower the COM for stability) [4]. The stepping strategy is used to establish a new BOS when the feet-in-place strategy is not enough to overcome the increasing perturbation intensity [4,5]. Compared to young adults, older adults tended to rely more on the proximal lower-limb joint motions and muscles than the distal ones, and may use the stepping strategy for reactive/compensatory/automatic balance control following unpredictable perturbations [6]. Apart from the age-related changes in the responses of multiple muscles/joints, prior studies have also shown the interaction effects of age with the perturbation direction and perturbation intensity on balance control strategies [6,7,8]. Nevertheless, the identified age-related kinematic and neuromuscular alterations underlying reactive balance control may not be directly indicative of fall risks, due to the potential existence of the confounding factor of age. Specific investigations and comparisons of the older adults with and without fall histories (i.e., fallers vs. non-fallers, and excluding the confounding factor of age) are therefore warranted to identify further balance control alterations in older individuals who are prone to falls, and to identify the fall-related factors.

Previous studies have intensively analyzed the stepping strategies and whole-body postural sways to compare the reactive balance control between fallers vs. non-fallers [9,10,11,12,13,14,15], while there has been less focus on the differences in specific joint motions or muscle activities [16,17,18]. Firstly, lower-limb muscle activities during reactive balance control have primarily been examined within a restricted number of lower-limb muscles, i.e., the ankle dorsiflexor/plantarflexor [16,17,18], knee flexor/extensor [16,18], and hip abductor [18]. The differences in hip adductor and hip flexor/extensor activation between fallers and non-fallers remain unknown. Secondly, prior investigations of lower-limb muscle activities have examined only one single EMG parameter in each study [16,17,18]. Fallers were reported to exhibit longer EMG onset latency of the ankle dorsiflexor following anterior translational perturbations during standing [17], longer EMG onset latencies of the hip abductor and knee flexor in the weight-bearing leg following lateral shoulder-impact perturbations during standing [18], and no significantly different agonist-antagonist co-contraction index (CCI) of the ankle dorsiflexor-plantarflexor or knee flexor-extensor following optical flow perturbations during walking as compared to non-fallers [16]. The existing analysis of timing and amplitude characteristics of EMG signals may have been insufficient, since only the delayed muscular reaction was identified to differentiate fallers from non-fallers [17,18]. Thirdly, regarding joint kinematics, interestingly, no prior studies seemed to have compared them in fallers vs. non-fallers during reactive balance control, to the best of authors’ knowledge. Although fallers exhibited a decreased range of motion in lower-limb joints than non-fallers [19], it has been unclear whether the lower-limb joint motions during reactive balance control differ between fallers and non-fallers or not. More comprehensive analyses of lower-limb muscle activities and joint kinematics are needed to facilitate the understanding of older fallers’ balance control strategies.

Reactive balance control strategies are influenced by both the perturbation direction and perturbation intensity [5,20], while there is still insufficient evidence on how fallers and non-fallers respond differently to diverse directions or intensities of balance perturbations. Regarding the perturbation intensity, a previous study reported that fallers and non-fallers’ differences in stepping strategy were more pronounced following a higher intensity of mediolateral perturbation [10], whereas another study did not observe an interaction effect of fall history and perturbation intensity on the reactive stepping strategy following unpredictable anterior perturbations [21]. Regarding the perturbation direction, prior studies also reported inconsistent differences in postural sway between fallers and non-fallers when responding to the unpredictable anteroposterior [13,21] or mediolateral [9,13,18,22] perturbations. The underlying reasons for these inconsistent findings have not been thoroughly understood/explained. Analyzing neuromuscular responses and joint kinematics during reactive balance control can potentially help better explain how the fall-prone older adults respond to varied levels of threats of suddenly losing balance. This may also provide useful insights for clinical assessments or even interventional enhancement of reactive balance.

The main aim of this study was therefore to explore the older fallers’ neuromuscular and kinematic alterations of lower limbs during reactive balance control as compared to non-fallers. Specifically, this study had the research question of how EMG/angle signals varied among the eight different dominant-leg muscles/joint motions, different fall histories, directions, and intensities of unpredictable translational perturbations. In addition, the study investigated how the COM displacements varied among the six different postural sway directions (i.e., forward/backward, medial/lateral, and upward/downward), different fall histories, perturbation directions, and perturbation intensities. The temporal parameters including onset latency, time to peak, and burst duration, together with the amplitude parameters including rate of rise, peak amplitude, and/or agonist-antagonist CCI were analyzed for these signals. We hypothesized that the analyzed parameters during reactive balance control would be affected by the interaction of fall history, muscle/joint motion/postural sway direction, perturbation direction, and perturbation intensity. Furthermore, for the effects of fall history, based on the previously available findings related to aging [4,6,7,16,19,23] and fall histories [16,17,18], we extrapolated that fallers would have delayed timing and larger amplitudes of proximal muscles’ activation/joint motions as compared to non-fallers following a high intensity of unpredictable anterior or lateral balance perturbation.

## 2. Materials and Methods

### 2.1. Study Design and Participants

This study was a pilot observational cross-sectional study. Participants were recruited through convenience sampling. Inclusion criteria were: (1) aged 65 years old or over, (2) living in the community independently and able to walk for 400 m without any assistance, and (3) fallers (with at least one fall within the past one year) or non-fallers (with no fall within the past one year), matched for age and sex. Exclusion criteria were: (1) being hospitalized or living in a nursing home for more than six months in the past year; (2) experiencing fall(s) due to traffic or occupational accidents; (3) being diagnosed with cognitive impairment or severe systemic disease (e.g., neuromuscular, renal, hepatic, orthopedic, vestibular, or cardiopulmonary disorders) that impacts or limits physical activities; and (4) participated in any structured exercise training or strengthening exercises within the past one year. A total of twelve older participants were finally eligible for this study. Before being tested, each participant read and signed an informed consent form to participate in this study (Ethics approval agency: Institutional Review Board, The Hong Kong Polytechnic University; Ethical reference number: HSEARS20201230002). Each participant underwent the experiment once, involving subjective assessments and perturbation trials.

### 2.2. Subjective Assessments

Collection of demographic data (e.g., age, sex, height, body mass), medical history, and fall history was first conducted, followed by assessments using questionnaires/scales. A fall was defined as an event that resulted in coming to rest inadvertently on the ground or floor or other lower level and not resulting from an intrinsic or overwhelming hazard [24]. The short Falls Efficacy Scale-International (FES-I) and the Chinese version of the Physical Activity Scale for the Elderly (PASE-C) were used for the measurement of a participant’s fear of falling and physical activity level, respectively [25,26]. The Mini-Balance Evaluation System Test (Mini-BEST) was used to assess a participant’s functional balance performance, including anticipatory postural control, reactive postural control, sensory orientation, and dynamic gait [27]. The Mini-BEST was selected due to its established reliability and validity [28] together with its comprehensive evaluation of various balance dimensions, particularly for the sub-item of reactive balance, making it more relevant to the topic of this study than other clinical assessments (e.g., Timed Up and Go test, Berg Balance Scale). Then the participant’s dominant leg was determined through the use of three tests: the balance recovery test, the ball-kick test, and the step-up test [5,29]. The leg that was used more frequently for stepping after being nudged forward, kicking a ball, and stepping onto a stair across a total of nine trials (three trials for each test) was identified as the dominant leg [5,29]. All subjective assessments were conducted by the same examiner.

### 2.3. Perturbation Trials

#### 2.3.1. Experimental Set-Up

A moving-platform perturbation system was used to induce the unpredictable translational perturbations (Figure 1), with technical details reported in a previous study [5]. Briefly, the platform can move horizontally at a random starting time, with random moving direction and random moving distance/velocity/acceleration (related to different intensities) to constitute an unpredictable balance perturbation to the participant standing on it. The whole-body kinematics were collected using an 8-camera motion capture system (Nexus 2.11, Vicon Motion Systems Ltd., Yarnton, UK) that sampled at 250 Hz. An eight-channel Trigno Wireless Biofeedback System (Delsys Inc., Natick, MA, USA) that sampled at 2000 Hz was used to record muscular electrical activities. The data collection was synchronized for the three systems [5].

#### 2.3.2. Protocol of Perturbation Trials

The procedure for the perturbation trials was firstly explained to the participant. Participants were asked in advance to wear their daily footwear, except for impractical shoes, such as sandals, high heels, ballet shoes and slippers. Each participant was given an identical set of tight shirt and shorts, to optimize the Vicon motion capture and the placement of retroreflective markers and EMG sensors. Before the perturbation trials, EMG sensors and retroreflective markers were placed on the participant. The eight wireless surface EMG sensors were placed on the eight dominant-leg muscles, according to the recommendation of the Surface ElectroMyoGraphy for the Non-Invasive Assessment of Muscles (SENIAM) project (Table A1 in Appendix B) [30]. The major muscles relevant to eight ankle, knee and hip joint motions were selected, including the tibialis anterior (TA), gastrocnemius medialis (GM), rectus femoris (RF), long head of bicep femoris (BF), sartorius (SA), gluteus maximus (GMax), gluteus medius (GMed), and adductor magnus (AM). A standard skin preparation procedure, including shaving, cleaning, and slightly abrading with alcohol wipes, was performed before attaching the EMG electrodes. The sensors were applied to the skin with double-sided tape (Trigno Sensor Adhesive Interface, Delsys, Boston, MA, USA), with medical tape to enhance fixation. Then a set of 39 retroreflective markers were attached to the bony landmarks of the head, torso, left and right upper limbs, pelvis, and left and right lower limbs [31]. All placements were conducted by the same examiner.

The participant was then instructed to stand with two feet wearing shoes and shoulder-width apart on the middle of the platform, holding a light rod at waist level and close to the trunk to keep the arms from blocking the reflective markers. The participant was told to stand naturally and look forward at the beginning, try best to maintain balance if feeling the perturbation, and then return to the original foot position marked by the dark-colored tapes as quickly as possible if the foot moved. A safety harness system (PG-360, Physio Gait Dynamic Unweighting System, Healthcare International Ltd., Langley, WA, USA) was equipped on each participant as a safety measure during the perturbation.

Each participant then experienced four trials (each consisted of 12 random perturbations) covering a total of 48 unpredictable balance perturbations (4 directions × 4 intensities × 3 repetitions), with 5 min of rest after each trial. The platform moved horizontally in a pre-determined direction and intensity first, then remained stationary for 12 s, and was finally pulled back to its original position. The triggering time, directions (anterior, posterior, medial, and lateral), and intensities (highest, high, low, and lowest) were randomized and blinded to the participant. The anterior, posterior, medial, and lateral perturbations induced the participant’s backward, forward, lateral, and medial loss of balance, respectively. Based on the human’s limits of stability in different directions and our pilot study results in young adults [5], the highest intensity for the anterior, posterior, medial, and lateral directions corresponded to the platform’s moving distances of 2.67%, 4.00%, 5.33% and 5.33% of each participant’s height, respectively. Based on the trajectory of the reflective marker placed on the platform, the displacement, velocity, and acceleration of platform movement for each intensity are presented in Table A2 of Appendix B. Videos were recorded in real time during all perturbation trials to enable further manual observation and analysis of balance control strategies. Regardless of whether feet-in-place strategies or stepping strategies were used, the dominant-leg muscle activities and joint motions were focused on and analyzed for all participants.

### 2.4. Data Processing

Kinematic data, including the whole body’s COM, the dominant-leg hip, knee, and ankle joint motions were first processed using the Plug-in Gait full body model. Then the kinematic data and the raw EMG data of dominant-leg muscles were further processed as described below in a custom MATLAB program MATLAB 2019b, The MathWorks, Inc., Natick, MA, USA). The kinematic data were subtracted by the mean signal value of the 1000-ms baseline interval before the start of each perturbation for normalization. To obtain the COM displacement relative to the base of support (BOS), the COM displacement was further subtracted by the displacement of the moving platform [5]. The raw EMG signals were zeroed to the mean value of the entire perturbation trial [32,33], full-wave rectified, and low-pass filtered at 4 Hz with a bi-directional 4th order Butterworth filer to obtain the envelope [32], then further divided by the mean signal value of the 1000-ms baseline interval before the start of the perturbation trial for normalization [5,20].

Temporal parameters, including the onset latency, time to peak, and burst duration, together with amplitude parameters, including the peak amplitude, rate of rise, and/or agonist-antagonist CCI were analyzed through a custom MATLAB algorithm (Figure 2). Within 2 s after the start of each perturbation, the onset was detected as the first point in time when the corresponding signal value exceeded five times the standard deviation (SD) over the mean baseline value (mean + 5 SD), and the peak was identified as the point with the maximum signal value after the onset [5,20,34]. The reason for detecting onset and peak within 2 s after a perturbation was because participants were observed to recover balance within this duration following perturbations in the pilot study, and the kinematic or EMG reactions within this duration were considered meaningful to resist the sudden balance loss. Within 9 s after the start of each perturbation, the offset was identified as the first point in time after the onset, when the corresponding signal value dropped below five times the standard deviation over the mean baseline value (mean + 5 SD) [35]. The baseline for onset or offset detection was the 1000-ms interval of a signal before the start of each perturbation. The onset latency indicated the time delay from the start of perturbation to the signal onset, the time to peak indicated that from the start of perturbation to the signal peak, and the burst duration indicated that from the signal onset to offset. The rate of rise was determined as the gradient of the signal rise within a 50-ms period following the onset [5,20]. The agonist-antagonist CCI within the duration from two muscles’ later EMG onset to two muscles’ earlier EMG offset was calculated based on the formula in Figure 2 [16,36,37]. When the onset of a signal was absent within 2 s after a perturbation, the onset latency, time to peak, and burst duration were filled with 2000 ms, 2000 ms, and 0 ms, respectively; while the peak amplitude, rate of rise, and agonis-antagonist CCI were all filled with 0. For each parameter, the mean value of the three perturbations with the same direction and intensity was used in further statistical analyses.

### 2.5. Statistical Analyses

The statistical analyses were performed using SPSS (version 25.0), with the significance level set as 0.05. To examine differences in baseline subjective assessment data between fallers and non-fallers, independent sample t tests or Mann-Whitney U tests were used based on the data normality for continuous variables, and Chi-square tests were used for categorical variables. For each parameter (i.e., onset latency, time to peak, peak amplitude, burst duration, peak amplitude, rate of rise, and/or agonist-antagonist CCI), a four-way analysis of variance (ANOVA) and post hoc pairwise comparisons with Bonferroni corrections were conducted to examine the effects of two fall statuses, four perturbation directions, four perturbation intensities, and six postural sway directions/eight dominant-leg joint motions/eight dominant-leg muscles/four dominant-leg muscle pairs. With samples of equal size, the ANOVAs were considered robust, even when the assumptions of normality and homogeneity were not fully met [38]. Given the objectives, this study focused on the interaction effects of fall history with other factor(s), the main effects of fall history when there were no significant interactions, and the simple effects of fall history when there were significant interactions. Results regarding the simple effects of the factor(s) that had interactions with fall history are presented in Table A3, Table A4 and Table A5 of Appendix B.

## 3. Results

### 3.1. Subjective Assessment Results

No adverse incidents happened during any of the experiments. In the faller group, two participants experienced recurrent falls and four had a single fall in the past year before the perturbation trials. There was no significant difference in the number of medications, age, body mass, height, foot length, BMI, short FES-I score or the PASE-C score between the participating older fallers and older non-fallers (Table 1). Nevertheless, the Mini-BEST score of fallers was significantly lower than that of non-fallers (*p* < 0.05).

### 3.2. Balance Control Strategies

Fallers were more likely to have stepping responses than non-fallers. The unpredictable translational perturbations mainly induced feet-in-place strategies (567/576, 98.4%), and three participants (3/12, 25.0%) showed stepping responses following nine perturbations (9/576, 1.6%). Specifically, following three highest-intensity medial perturbations, one non-faller exhibited responses of the non-dominant leg, including stepping toward the perturbation direction, performing leg abduction, and elevating the leg (3/576, 0.5%). One faller took a backward step using the non-dominant leg together with several small steps, following a highest-intensity anterior perturbation (1/576, 0.2%). The other faller stepped backward using the non-dominant leg, following the highest-intensity (2/576, 0.3%) and high-intensity (1/576, 0.2%) anterior perturbations. Additionally, this individual stepped toward the perturbation direction using both legs in response to a low-intensity posterior perturbation (1/576, 0.2%), and with the non-dominant leg in response to a highest-intensity medial perturbation (1/576, 0.2%).

### 3.3. COM Displacements

The mean changes in COM displacements over time (*n* = 12, Figure 3) together with the onset latency, time to peak, peak amplitude, and burst duration of COM displacement (mean ± SD; Figure 4) are displayed for each postural sway direction, each perturbation intensity, and each perturbation direction in participating older fallers and older non-fallers.

Regarding the onset latency of COM displacement, there was a significant interaction between “fall history” and “postural sway direction” factors (*p* < 0.05). Significant simple effects of fall history showed that fallers had a longer onset latency in the backward direction, but had shorter onset latencies in the forward and downward directions compared to non-fallers (*p* < 0.05, Figure 4).

Regarding the time to peak COM displacement, a significant interaction was also found between “fall history” and “postural sway direction” factors (*p* < 0.05). Significant simple effects of fall history showed that fallers had a longer time to peak COM displacement in the backward direction, but had a shorter time to peak in the downward direction compared to non-fallers (*p* < 0.05; Figure 4).

Regarding the peak COM displacement, a significant interaction of fall history with other factors was observed (fall history × direction × postural sway direction, *p* < 0.05). Significant simple effects of fall history showed that fallers had larger peak COM displacements in the forward and downward directions following anterior perturbations, in the backward direction following posterior perturbations, and in the forward direction following both medial and lateral perturbations (*p* < 0.05; Figure 4).

Regarding the burst duration of COM displacement, there was no significant interaction of fall history with any other factor. No significant main effect of fall history was observed on this parameter either (Figure 4).

### 3.4. Dominant-Leg Joint Motions

The mean changes of dominant-leg joint motions over time (Figure 5) together with the angle onset latency, time to peak angle, peak angle, and angle burst duration (mean ± SD; Figure 6) are displayed for each joint motion, each perturbation intensity, and each perturbation direction in fallers and non-fallers.

Regarding the angle onset latency, a significant interaction of fall history with other factors was observed (fall history × direction × joint motion, *p* < 0.05). Significant simple effects of fall history showed that compared to non-fallers, fallers had longer onset latencies of hip adduction, hip extension, and knee extension following anterior perturbations, and a longer onset latency of ankle plantarflexion following medial perturbations (*p* < 0.05; Figure 6).

Regarding the time to peak angle, a significant interaction of fall history with other factors was also observed (fall history × direction × joint motion, *p* < 0.05). Significant simple effects of fall history showed that fallers had longer time to peak hip adduction, hip flexion, hip extension, and knee extension following anterior perturbations, longer time to peak ankle plantarflexion following medial perturbations, but shorter time to peak ankle plantarflexion following lateral perturbations as compared to non-fallers (*p* < 0.05; Figure 6).

Regarding the peak angle, there was a significant interaction between “fall history” and “joint motion” factors (*p* < 0.05). Significant simple effects of fall history showed that fallers had larger peak hip flexion and knee flexion than non-fallers following unpredictable perturbations (*p* < 0.05; Figure 6).

Regarding the angle burst duration, a significant interaction was also observed between “fall history” and “joint motion” factors (*p* < 0.05). The significant simple effect of fall history showed that fallers had a longer burst duration of ankle dorsiflexion than non-fallers following unpredictable perturbations (*p* < 0.05; Figure 6).

### 3.5. EMG Signals of Dominant-Leg Muscles

The mean changes of EMG signals over time (Figure 7) together with the EMG onset latency, rate of EMG rise, time to peak EMG amplitude, EMG burst duration, and agonist-antagonist CCI (mean ± SD; Figure 8) are presented for each dominant-leg muscle (pair), each perturbation intensity, and each perturbation direction in fallers and non-fallers.

Regarding the EMG onset latency, there was a significant interaction between “fall history” and “muscle” factors (*p* < 0.05). Significant simple effects of fall history showed that the fallers’ EMG onset latencies were longer for the hip flexor and hip extensor compared to non-fallers following unpredictable perturbations (*p* < 0.05; Figure 8).

Regarding the rate of EMG rise, there was also a significant interaction between “fall history” and “muscle” factors (*p* < 0.05). Significant simple effects of fall history showed that the fallers’ rate of EMG rise was smaller for the ankle dorsiflexor, but was larger for the knee flexor as compared to non-fallers following unpredictable perturbations (*p* < 0.05; Figure 8).

Regarding the time to peak EMG amplitude, there was no significant interaction of fall history with any other factor. The significant main effect of fall history showed that fallers had longer time to peak EMG amplitudes of lower-limb muscles as compared to non-fallers following unpredictable perturbations (*p* < 0.05; Figure 8).

Regarding the EMG burst duration, a significant interaction between “fall history” and “muscle” factors was observed (*p* < 0.05). Significant simple effects of fall history showed that fallers had longer EMG burst durations of the hip abductor and ankle dorsiflexor, but had a shorter one of the hip flexor than non-fallers following unpredictable perturbations (*p* < 0.05; Figure 8). In addition, there was a significant interaction between “fall history” and “direction” factors (*p* < 0.05). Significant simple effects of fall history showed that fallers had longer EMG burst durations following the anterior and posterior perturbations, but had a shorter one following the medial perturbations compared to non-fallers (*p* < 0.05; Figure 8).

Regarding the time to peak EMG amplitude, no significant interaction was observed between fall history and any other factor. The significant main effect of fall history showed that fallers had smaller agonist-antagonist CCIs in the investigated muscle pairs than non-fallers following unpredictable perturbations (*p* < 0.05; Figure 8).

## 4. Discussion

This study comprehensively examined the effects of fall history on reactive standing balance in community-dwelling older adults, by focusing on dominant-leg muscle activities and joint motions. Partially in line with our hypotheses, the effects of “fall history” on the investigated outcomes during reactive balance control interacted with the “muscle/joint motion/postural sway direction” and “perturbation direction” but not with the “perturbation intensity”. Specifically, compared to older non-fallers, older fallers demonstrated slowed activation of ankle/hip muscles while tending to use a suspensory strategy for reactive balance control, as supported by a series of neuromuscular alterations and joint kinematics. These new insights reflect possible reasons for fallers’ decreased balance capability and indicate their utilization of prolonged and enlarged (and even overreacted) compensatory strategies for preserving postural stability [19]. Developing some future assessment tools based on the identified parameters may be helpful to screen and identify the fallers from non-fallers in the community-dwelling adults. Furthermore, interventions targeting these identified fall-related alterations may lead to more effective solutions for improving reactive balance control and preventing recurrent falls in older fallers. Details are discussed below.

### 4.1. Fallers Tended to Use Suspensory Strategies Following Unpredictable Perturbations: Neuromuscular and Kinematic Mechanisms

The primary finding of this study was that fallers have tended to use the suspensory strategy to maintain standing balance following unpredictable translational perturbations as compared to non-fallers. This strategy enabled fallers to promptly compensate for their insufficient initiation of ankle and hip strategies, but it led to their prolonged and overacted balance recovery. This finding provides insights into the specific decline in reactive balance control and compensation strategies among the older adults with fall histories and poor overall balance performance (i.e., low Mini-BEST scores).

Fallers exhibited decreased activation in the ankle dorsiflexor and hip flexor/extensor as compared to non-fallers. This could be attributed to the potential degradation in any components along the sensorimotor pathway, including sensory input (feedback from external perturbation), central organization, and motor output [6]. Ankle dorsiflexor’s activation immediately following the start of perturbation has been in the first line to resist the sudden loss of balance [5]. With ageing, humans may shift from a distal-to-proximal strategy to a proximal-to-distal strategy to maintain balance, compensating for the difficulties of generating sufficient ankle torque [39]. This study further proved that this phenomenon was more pronounced in older fallers than older non-fallers. On the other hand, fallers showed delayed EMG onset timing of the hip flexor/extensor and reduced EMG burst duration of the hip flexor as compared to non-fallers. These alterations could partly restrict the initiation of the hip strategy, which is the second line of defense against sudden loss of balance. The delayed activation of hip muscles may be explained by previous morphological observations, where fallers were found to have reduced density of skeletal muscle fibers and increased intramuscular adipose issues in the gluteus muscles as compared to non-fallers [40]. A prior study also reported delayed neuromuscular activation in reactive standing balance, with fallers exhibiting later EMG onset timing of the hip abductor and knee flexor in the loading leg than non-fallers following unpredictable lateral perturbations exerted on the shoulder [18]. The discrepancy in the affected muscles could be attributed to the different perturbation methods.

A series of kinematic and neuromuscular alterations in fallers when facing unpredictable translational balance perturbations have indicated their prominent use of suspensory strategies as compared to non-fallers. In the absence of sufficient ankle and hip muscle activation, fallers utilized the suspensory strategy, i.e., the third strategy to resist sudden loss of balance, by lowering the COM to increase the limit of stability and absorb the external perturbation [41,42,43]. This is evident from their earlier onset and peak timing of downward COM displacement compared to non-fallers. The increased activation rate of the knee flexor, generally decreased agonist-antagonist co-contraction of the lower-limb muscles, and larger knee/hip flexion in fallers may have facilitated this strategy. Interestingly, our findings differ from a prior study that reported no differences in postural sway timing or amplitude between fallers and non-fallers following lateral shoulder-impact perturbations [18], suggesting that different body segment perturbations may elicit distinct reactive balance control strategies. Additionally, while previous research linked greater co-contraction to more joint stability and poorer balance control [44,45], this study observed that older fallers were able to reduce agonist-antagonist co-contractions of lower-limb muscles and achieve larger knee/hip flexion for a suspensory strategy. On top of them, this study has observed that fallers showed longer activation durations of the ankle dorsiflexor and hip abductor, together with a longer ankle dorsiflexion duration than non-fallers, which may be necessary for maintaining a knee bending posture during the suspensory strategy.

Fallers’ balance control strategies in the current study, however, have required a prolonged recovery time and caused overreactions. This is evidenced by their neuromuscular and kinematic alterations, as described below. Firstly, fallers’ delayed time to peak activation may suggest reduced motor unit recruitment and firing rate in response to external perturbations [23]. Secondly, fallers showed longer time to peak hip flexion angle following anterior perturbations, longer burst durations of ankle dorsiflexion following all perturbations, and delayed timing of recovery joint motions following anterior/medial perturbations than non-fallers. Thirdly, fallers showed larger overshooting postural sways when recovering to initial positions following unpredictable anteroposterior perturbations. This is indicated by their larger forward peak COM displacements following sudden backward balance loss (induced by anterior perturbations) and larger backward ones following sudden forward balance loss (induced by posterior perturbations) as compared to non-fallers. These findings indicate that sudden perturbations could pose greater challenges to older fallers. Fallers’ more prominent overshoots of backward postural sways, as compared to non-fallers, have also been previously reported following anterior waist-pull perturbations [46]. Additionally, a prior study found that fallers had more variable and delayed recovery steps than non-fallers in perturbed walking [11], which could be corroborated by this study’s observation of fallers’ larger overshooting postural sways and delayed timing of overshooting lower-limb joint motions. The slowed but exaggerated postural adjustments seem to reveal the ineffective strategies used by the older adults with fall histories for reactive balance control, aligning with their poorer overall balance performance, as shown by the lower Mini-BEST scores.

This study also observed large within-group variations in some of the analyzed parameters during reactive balance control. For example, although the faller group was identified to exhibit longer EMG onset latencies in hip muscles than the non-faller group, the within-group variations were large, as indicated by the standard deviations. This reflects the fact that even individuals with the same fall status may respond differently during reactive balance control. On the one hand, a previous study reported that fallers had a higher variation in walking stability than non-fallers after unpredictable perturbations [11]. Given its potential in differentiating fall status, the variability of reactive balance performance should not be neglected. On the other hand, analysis of individual variations in reactive balance control within the faller group (e.g., coefficient of variation) could be also crucial for understanding personalized balance intervention requirements, which merits further investigation.

### 4.2. Fallers Had Altered Responses to Different Perturbation Directions

The secondary finding of this study was that fall history showed interaction effects with perturbation direction, but not with perturbation intensity, on the older adults’ neuromuscular and kinematic responses during reactive balance control.

Fallers have shown distinct responses to anteroposterior and mediolateral perturbations as compared to non-fallers. Regarding kinematics, a previous study reported fallers’ larger COM path displacement as compared to non-fallers following the mediolateral translational perturbations [9]. This study has further revealed that fallers’ larger postural sways were specifically in the forward direction following the mediolateral perturbations. These responses could be partly attributed to fallers’ delayed timing of recovery joint motions compared to non-fallers following the medial perturbation. Regarding the neuromuscular responses, fallers have exhibited longer EMG burst durations in dominant-leg muscles following anteroposterior perturbations as compared to non-fallers. This could also explain fallers’ overshooting postural sways. Additionally, this could explain why fallers had more non-dominant leg stepping following anteroposterior perturbations than non-fallers, as more body weight was loaded on the dominant leg. Conversely, fallers have exhibited shorter EMG burst durations of dominant-leg muscles than non-fallers following medial perturbations, resulting in fallers’ fewer non-dominant leg steps following medial perturbations compared to non-fallers.

Notably, this study has found no differences in the responses of fallers compared to non-fallers to varied intensities of unpredictable perturbations. Previous studies reported inconsistent results regarding the interaction effect of fall history and perturbation intensity on the stepping characteristics following waist-pull perturbations [10,21]. The finding of this study has further built on the evidence of unpredictable translational perturbations and suggested that fallers’ neuromuscular/kinematic responses to the different intensities of perturbations, which primarily induced feet-in-place strategies, were similar to those of non-fallers.

### 4.3. Strengths and Limitations

To the best of authors’ knowledge, this study offers a preliminary but more in-depth exploration of the differences in eight major lower-limb muscles’ activation or lower-limb joint kinematics during reactive balance control between older fallers and older non-fallers. With the comprehensive analyses of temporal and amplitude characteristics of these investigated signals, this study has built knowledge upon the prior investigations that focused on a limited number of muscles and EMG parameters, and has addressed the gap of limited research on joint kinematics in fallers vs. non-fallers. The mechanisms of fall-prone older adults’ decline of reactive balance control and compensatory strategies could be better understood with the findings of this study.

This study has several limitations that shall be acknowledged, and the findings shall be interpreted with caution. (1) Given the small sample sizes of recruited older fallers and older non-fallers, it should be noted that the findings of this pilot study using four-way ANOVAs can provide only preliminary insights. Results of alternative analyses for the “fall history” factor, using independent sample t-tests for normally distributed data and Mann-Whitney U tests for non-normally distributed data, are presented in Appendix A. In addition, this study did not specifically analyze the recurrent fallers’ neuromuscular/kinematic strategies in reactive balance control, which demands further studies with larger sample sizes. (2) This study did not ensure that the faller and non-faller groups had matched ranges of motion in the investigated joints or muscle strength, which may confound the identified between-group differences in reactive balance performance. Additionally, this study only focused on reactive balance control in fall-prone people. It is important to note that the causes of loss of balance or falls are not confined to this and are multi-factorial (e.g., environmental factors). (3) Although diagnosis of cognitive impairment was an exclusion criterion for this study and the testers ensured participants understood task instructions, a clinical scale such as the Montreal Cognitive Assessment (MoCA) should be used to better quantify the older participants’ cognitive function in future research. (4) The cut-off frequency used for EMG low-pass filtering in this study may not be optimal for all of the investigated eight leg muscles. Further attempts are warranted to determine the cut-off frequency for processing each respective muscle’s EMG signal based on the frequency spectrum analysis. Additionally, this study did not conduct maximal voluntary contraction tests but used the baseline EMG signal value in unperturbed standing for EMG amplitude normalization. It is therefore important to note that the rate of EMG rise and agonist-antagonist CCI in this study reflected the extent to which the perturbation task utilized the activation required for normal standing rather than the maximal activation. (5) This study did not explore gender-specific variations in balance control by having equal representation of male and female participants in a group. Future studies could consider specifically examining the gender-specific difference in older people with a larger sample size to address this issue.

### 4.4. Implications for Clinical Practice and Future Resesarch

This study has preliminary implications for assessing and training reactive balance control in future applications. On the one hand, the identified fall-related kinematic and neuromuscular factors may inform clinical practice. Reactive balance training may need to be prescribed more for the community-dwelling older adults with fall histories in the future, considering their generally delayed peak activation of lower-limb muscles. A recent review has reported that perturbation-based balance training and stepping training can improve the slowed reaction time in response to sudden loss of balance [47]. This is a promising future avenue, as fallers’ degradation in neuromuscular timing can be modified. Our findings further imply that more focus/efforts may need to be put into the ankle and hip muscle power (or reaction speed) training in older fallers to improve their reactive balance control performance. Although the proximal hip and knee muscles were previously reported to be more affected by aging following unpredictable perturbations [48], the findings of this study may suggest that the training of ankle muscles should not be ignored, especially in older fallers.

On the other hand, with the advancement of wearable sensors and real-time monitoring systems, future studies could also consider employing these tools to improve balance assessment and training in fall-prone individuals. Real-time monitoring and analysis of the reaction speed (e.g., EMG onset latency, rate of EMG rise) of an older client’s ankle dorsiflexor or hip flexor/extensor for reactive balance control may potentially enhance the fall-risk assessment on top of the current reactive balance test. The EMG-based biofeedback may potentially enhance power training for ankle or hip muscles, while the quantitative results on muscle reaction speed may help therapists offer more personalized feedback and guidance during reactive balance training.

Nonetheless, we acknowledge that these implications need validation in future longitudinal studies with larger sample sizes, where the diagnostic accuracy of slower activation in ankle and hip muscles in differentiating older adults’ fall risks, as identified in this pilot study, should be examined.

## 5. Conclusions

This pilot study found that older fallers’ kinematic and neuromuscular alterations in resisting unpredictable translational perturbations could be indicators of both the decline and the compensation of reactive balance control. Compared to non-fallers, fallers had a decreased activation rate in the ankle dorsiflexor and delayed activation in the hip flexor/extensor, thereby resorting to the suspensory strategy for quickly responding to external perturbations. The increased activation rate of knee flexor, decreased agonist-antagonist co-contraction of lower-limb muscles, enlarged knee and hip flexion, and earlier downward postural sways in fallers could be the basis of their prioritizing of suspensory strategies as compared to non-fallers. However, fallers’ balance control strategies required prolonged recovery in lower-limb joint motions and caused overreactions in postural sways. A further longitudinal study with a larger sample is merited to verify these fall-related factors, which could enhance the identification of fall-prone people and provide insights for more targeted fall-prevention strategies.

## Figures and Tables

**Figure 1 bioengineering-12-00066-f001:**
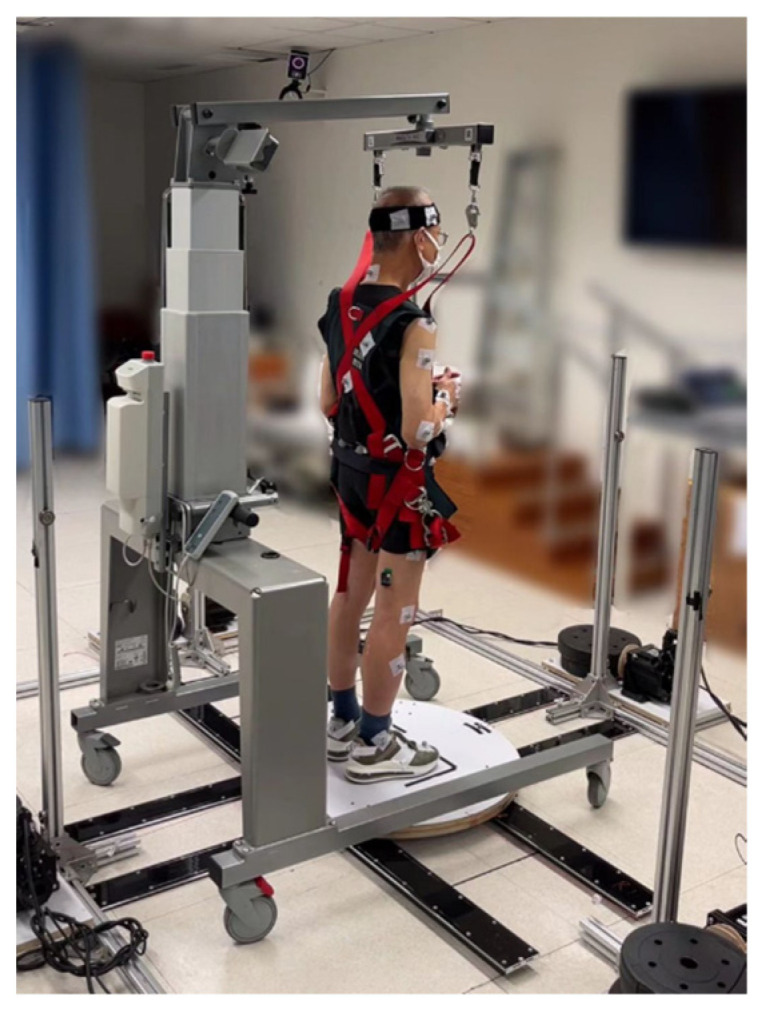
The moving-platform perturbation system being used by a participant.

**Figure 2 bioengineering-12-00066-f002:**
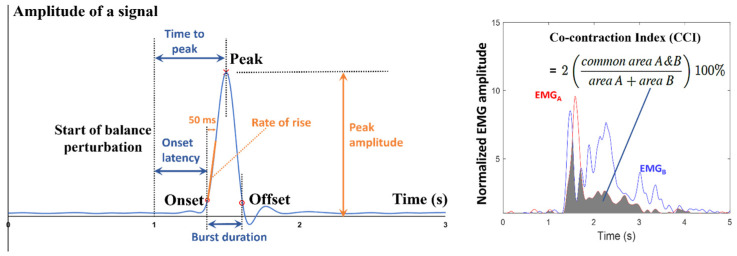
Illustrations of the analyzed temporal and amplitude parameters. **EMG**: electromyographic.

**Figure 3 bioengineering-12-00066-f003:**
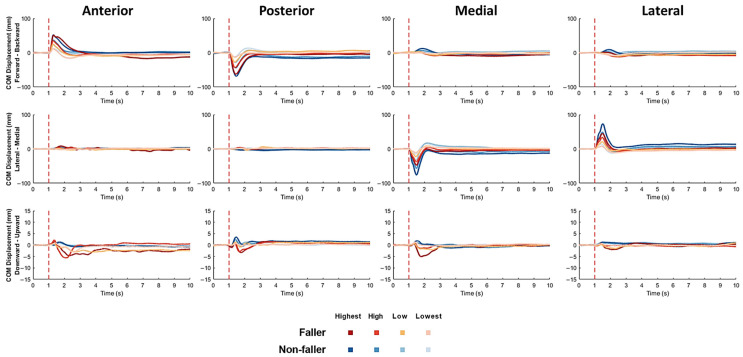
Mean forward/backward, medial/lateral, and upward/downward COM displacements in fallers (*n* = 6) and non-fallers (*n* = 6) following perturbations with different directions and intensities. The red dotted line denotes the start of balance perturbation. **COM**: center of mass.

**Figure 4 bioengineering-12-00066-f004:**
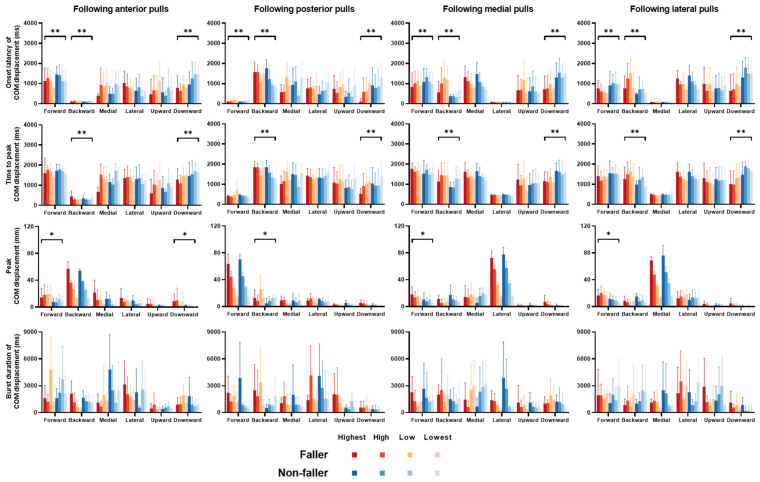
Onset latencies, time to peak, peak amplitudes, and burst durations of COM displacements in six postural sway directions (*n* = 6, mean ± SD). Significant effects of fall history are indicated by the 
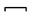
 (*p* < 0.05). ** denotes the significant effect of fall history at a certain postural sway direction (“fall history × postural sway direction” interaction). * denotes the significant effect of fall history at a certain postural sway direction and following a certain direction of perturbation (“fall history × postural sway direction × direction” interaction). **COM**: center of mass. **SD**: standard deviation.

**Figure 5 bioengineering-12-00066-f005:**
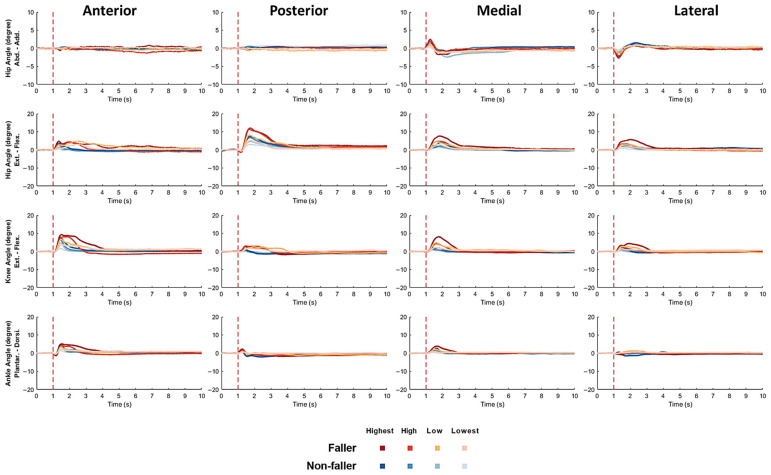
Mean change for each of the eight dominant-leg joint motions in fallers (*n* = 6) and non-fallers (*n* = 6) following perturbations with different directions and intensities. The red dotted line denotes the start of balance perturbation. **Add**: adduction. **Abd**: abduction. **Flex**: flexion. **Ext**: extension. **Dorsi**: dorsiflexion. **Plantar**: plantarflexion.

**Figure 6 bioengineering-12-00066-f006:**
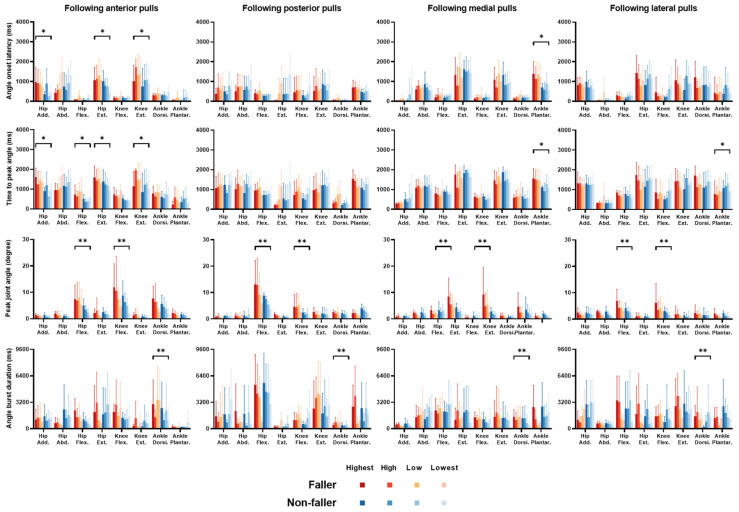
Onset latencies, time to peak, peak amplitudes, and burst durations of hip, knee, and ankle joint motions (*n* = 6, mean ± SD). Significant effects of fall history are indicated by the 
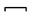
 (*p* < 0.05). * denotes the significant effect of fall history at a certain joint motion and following a certain direction of perturbation (“fall history × joint motion × direction” interaction). ** denotes the significant effect of fall history at a certain joint motion (“fall history × joint motion” interaction). **Add**: adduction. **Abd**: abduction. **Flex**: flexion. **Ext**: extension. **Dorsi**: dorsiflexion. **Plantar**: plantarflexion. **SD**: standard deviation.

**Figure 7 bioengineering-12-00066-f007:**
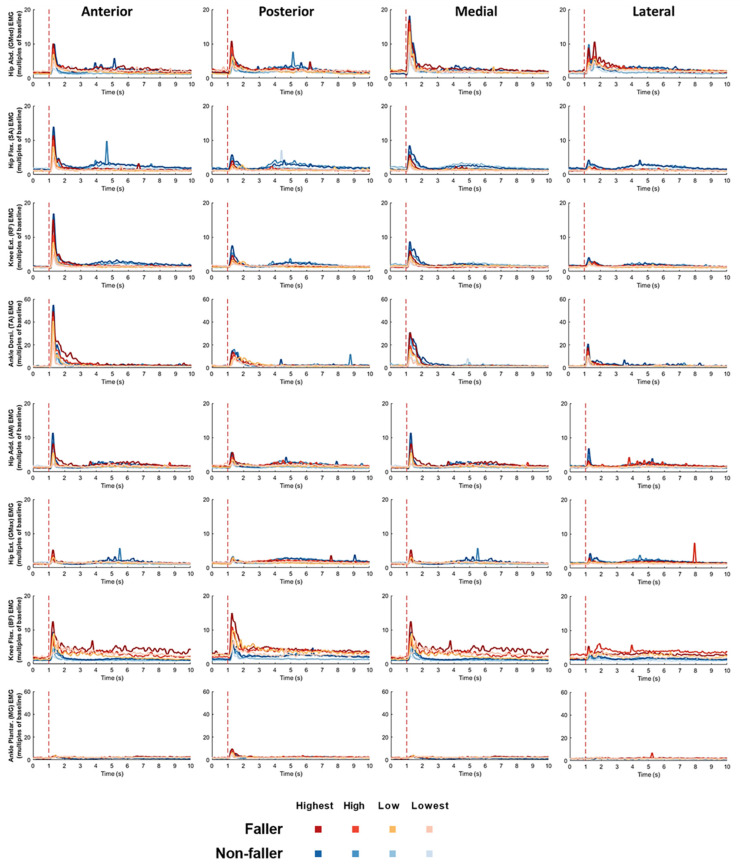
Mean EMG signal change for each of the eight dominant-leg muscles in fallers (*n* = 6) and non-fallers (*n* = 6) following perturbations with different directions and intensities. The red dotted line denotes the start of balance perturbation. **EMG**: electromyographic. **GMed**: gluteus medius. **SA**: sartorius. **RF**: rectus femoris. **TA**: tibialis anterior. **AM**: adductor magnus. **GMax**: gluteus maximus. **BF**: biceps femoris. **GM**: gastrocnemius medialis. **Add**: adductor. **Abd**: abductor. **Flex**: flexor. **Ext**: extensor. **Dorsi**: dorsiflexor. **Plantar**: plantarflexor.

**Figure 8 bioengineering-12-00066-f008:**
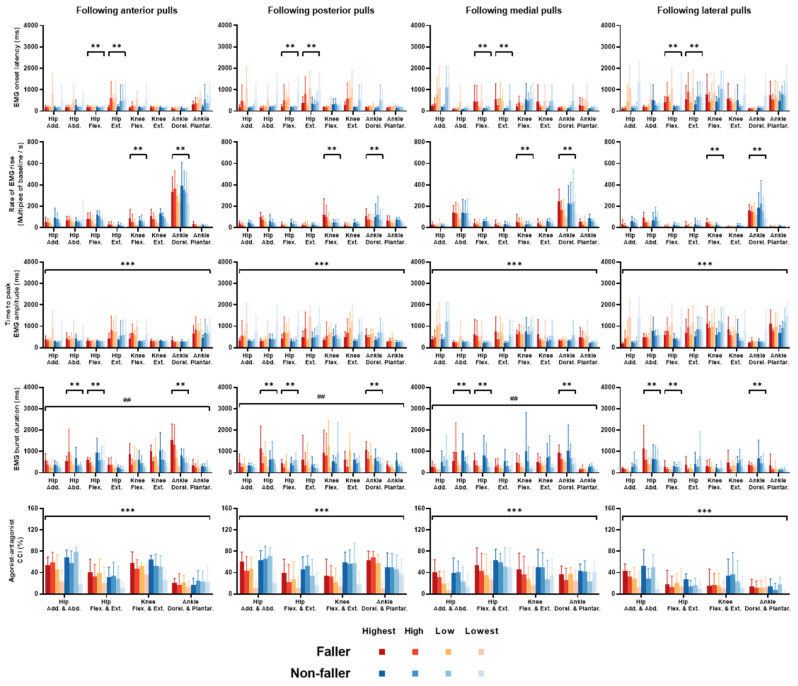
Onset latencies, rate of rise, time to peak, burst durations, and agonist-antagonist CCIs of EMG signals for eight dominant-leg muscles (*n* = 6, mean ± SD). Significant effects of fall history are indicated by the 
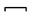
 (*p* < 0.05). *** denotes the significant main effect of fall history (no interaction of “fall history” and any other factor). ** denotes the significant effect of fall history at a certain muscle (“fall history × muscle” interaction). ## denotes the significant effect of fall history following a certain direction of perturbation (“fall history × direction” interaction). **EMG**: electromyographic. **CCI**: co-contraction index. **SD**: standard deviation. **Add**: adductor. **Abd**: abductor. **Flex**: flexor. **Ext**: extensor. **Dorsi**: dorsiflexor. **Plantar**: plantarflexor.

**Table 1 bioengineering-12-00066-t001:** Subjective assessment results (categorical variable: ratio; continuous variable: mean ± SD) of twelve participants.

	**Faller (*n* = 6, 3 Male** **s and 3 Female** **s)**	**Non-Faller (*n* = 6, 3 Male** **s and 3 Female** **s)**	**Significance (*p* Value)**
Number of falls	1.3 ± 0.5	0	/
Number of medications	1.0 ± 1.1	0.3 ± 0.5	0.279
Age (year)	71.5 ± 4.6	69.2 ± 2.9	0.316
Body mass (kg)	55.6 ± 8.4	61.4 ± 13.0	0.381
Height (cm)	157.9 ± 8.7	162.0 ± 7.9	0.406
BMI (kg/m^2^)	22.2 ± 2.0	23.3 ± 4.4	0.587
Leg length (cm)	77.3 ± 6.3	80.8 ± 4.6	0.297
Dominant leg (right/left)	5/1	6/0	0.296
Short FES-I (score)	12.2 ± 2.4	11.5 ± 6.0	0.332
PASE-C (score)	139.5 ± 73.2	148.1 ± 34.6	0.802
Mini-BEST (score)	23.3 ± 1.5	26.0 ± 0.9	**0.004**

**BMI**: body mass index. **FES-I**: Falls Efficacy Scale-International. **PASE-C**: Physical Activity Scale of Elderly-Chinese. **Mini-BEST**: Mini-Balance Evaluation System Test.

## Data Availability

Data are contained within the article and Appendix B.

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
