# Peer review of "Older Fallers’ Comprehensive Neuromuscular and Kinematic Alterations in Reactive Balance Control: Indicators of Balance Decline or Compensation? A Pilot Study"

_bioengineering, 2025, doi:10.3390/bioengineering12010066_

Round 1

Reviewer 1 Report

Comments and Suggestions for Authors

The authors adopted a very high-dimensional analysis approach to the issue of postural response after perturbations for fallers and non-fallers. The major issues of this manuscript are:

1). Insufficient description of perturbation parameters (should also describe velocity and acceleration) and responses (for example: dominant or non-dominant leg stepping);

2). Not clearly stated if both stepping responses and in-place responses are included in the analysis. If stepping responses are included, then justifications are needed for why choosing the dominant leg for recording (subjects may step with non-dominant leg);

3).  For the EMG analysis, it is very unclear why 2000 ms & 0 ms are assigned as outcomes if there is no EMG onset after perturbation. What is the justification? What is the % of EMG data with no EMG onset? How does this assignment affect data distribution (make it a skewed distribution?) and subsequent statistical analysis?

4). Need to provide a rationale of why averaging responses across subjects is reasonable. The authors aim to describe response strategies and it is conceivable that individual subjects (even with the same fall status) may have different strategies. If one subject has a big response that can skew the results significantly.

5). The sample size is likely to be severely under-powered for a 4-way ANOVA analysis. Further, the statistical model is not fully described. Are all possible 3-way, 2-way interactions specified in the model? How many total pair-wise post-hoc comparisons? What is the adjusted p-value to infer statistically significant difference?

6). Statistical results should be presented with all significant interactions, and main effects. The authors did not present any information on the main effects; therefore, it is not justified to directly compare the difference between fallers and non-fallers.

Reviewer 2 Report

Comments and Suggestions for Authors

Major Issues:

1.     The study employs the Mini-BEST tool but does not compare it with other balance assessments, thereby limiting its effectiveness.

2.     The study's results, based on only twelve participants, may lack robustness, reducing generalizability and making it difficult to draw broader conclusions.

3.     The paper neglects to consider individual variations in neuromuscular response strategies, which could be crucial for comprehending personalized balance intervention requirements.

4.     The study's lack of longitudinal analysis restricts understanding of how balance control mechanisms and fall risk may change with recurrent falls or aging.

5.     The study's findings highlight fall-prone traits but do not propose a clear or novel clinical tool or assessment method for real-world application.

6.     The work did not consider cognitive factors such as attention and reaction speed, which are important for older adults to understand balance control and restore balance.

7.     The paper lacks innovative use of wearable sensors or real-time monitoring systems to improve the detection and assessment of balance issues in fall-prone individuals.

8.     EMG signal processing involves zeroing and filtering data using standard parameters, but may not be optimal for all muscles or perturbation conditions, potentially causing data loss.

9.     The study's results may be indirect due to the inability to consider other factors affecting balance and muscle activation, such as individual strength and pre-existing balance training.

10.  The study's single experimental setup for balance perturbation may limit the applicability of findings and potentially overlook other real-world factors affecting balance in older adults.

Minor issues:

1.     The study neglects to consider environmental factors such as surface types, footwear, and lighting, which are known to affect balance and fall risk.

2.     The study did not explore gender-specific variations in balance control by having equal representation of male and female participants in a group.

Reviewer 3 Report

Comments and Suggestions for Authors

The authors of the work 'Older Fallers and Non-fallers’ Neuromuscular and Kinematic Alterations in Reactive Balance Control: Indicators of Balance Decline or Compensation?' describe the experimental procedure and results of a work aimed at investigating the muscle and joint kinematics as a consequence of balance reaction after an unpredictable translational balance perturbation. The study was focused on older fallers and non-fallers in order to investigate similarities or differences in the reactions of both populations.

The study, with both the experimental arrangement and the statistical analysis of the results.  is correctly described, and the results are discussed in depth.  The only critical point of the study is the very short population (six fallers and six non-fallers) analyzed in the work.  The authors discuss this point in the section 4.3 of the paper, and consider that, due to that limitation the results in this study can be considered as preliminary.

In spite of that limitation, the study has been well developed, and the statistical analysis of the recorded data shows clear differences in the behavior of fallers and non-fallers.  The authors' discussion on implications and perspectives also opens several possibilities for improving the work using wearable sensors, and for using the results in this paper in recovering therapies for more personalized treatments.

Considering all this, I recommend the publication of this paper in its current form.

Round 2

Reviewer 2 Report

Comments and Suggestions for Authors

The revised manuscript has reached its level to be published. The issues that I raised in the my previous comments have been successfully addressed.